# Epiphytic Fungi Can Drive a Trade-Off Between Pathogen and Herbivore Resistance in Invasive *Ipomoea cairica*

**DOI:** 10.3390/microorganisms12112130

**Published:** 2024-10-24

**Authors:** Hua Xu, Lixing Wang, Minjie Zhu, Xuhui Chen

**Affiliations:** 1School of Modern Industry for Selenium Science and Engineering, Wuhan Polytechnic University, Wuhan 430048, China; xuhua04@bnuz.edu.cn (H.X.); wlx13453764240@163.com (L.W.); 2Hunan Polytechnic of Environment and Biology, Hengyang 421005, China; zhuminjie@mail.nankai.edu.cn; 3College of Bioscience and Biotechnology, Shenyang Agricultural University, Shenyang 110161, China

**Keywords:** epiphytic fungi, invasive plant, trade-offs, herbivore defense, pathogen defense, biocontrol strategy

## Abstract

Trade-offs between different defense traits exist commonly in plants. However, no evidence suggests that symbiotic microbes can drive a trade-off between plant pathogen and herbivore defense. The present study aims to investigate whether the mixture of epiphytic *Fusarium oxysporum* and *Fusarium fujikuroi* can drive the trade-off between the two defense traits in invasive *Ipomoea cairica*. Surface-sterilized *I. cairica* cuttings pre-inoculated with the epiphytic fungal mixture served as an epiphyte-inoculated (E+) group, while cuttings sprayed with sterile PDB served as an epiphyte-free (E−) group. After 3 days of incubation, E+ and E− cuttings were subjected to the challenge from a fungal pathogen and an insect herbivore, respectively. The results suggested that E+ cuttings had less rotted and yellowed leaf rates per plant than E− cuttings after *Colletotrichum gloeosporioides* infection. On the contrary, E+ cuttings had higher absolute and relative fresh weight losses per leaf than E− cuttings after *Taiwania circumdata* introduction. In the absence of challenges from the two natural enemies, salicylic acid and H_2_O_2_ accumulation occurred in E+ cuttings, which activated their SA-dependent pathogen defense and resulted in an increase in chitinase and β-1,3-glucanase activities. Although jasmonic acid accumulation also occurred in E+ cuttings, their JA-dependent herbivore defense responses were antagonized by SA signaling, leading to a decrease in total phenol content and phenylalanine ammonia-lyase activity. The activity of generalized defense enzymes, including superoxide dismutase, peroxidase, and catalase, did not differ between E+ and E− cuttings. Together, our findings indicate that a trade-off between pathogen and herbivore defense in *I. cairica* had already been driven by the epiphytic fungal mixture before the challenge by the two natural enemies. This study provides a novel insight into biocontrol strategies for *I. cairica*.

## 1. Introduction

To achieve optimal fitness under variable environmental conditions, including biotic and abiotic stress, plants need to adopt strategies to balance the relationship between different traits [1]. The balanced relationship between different plant traits is defined as a trade-off [2]. Plant trait trade-offs are based on the fitness costs incurred when a favorable change in one life history trait is coupled to a harmful change in another trait [3]. Thus, the trade-offs are regarded as the result of species evolution [4].

The allocation of limited resources between multiple plant traits, including growth, reproduction, and tolerance to biotic and biotic stress, is considered to be the basis of trade-offs [5]. Many factors, including resource availability (e.g., nutrients and water) [6], biotic and abiotic stress [7], and plant genotypes, can influence allocation patterns of resources between different plant traits [8,9]. Interestingly, beneficial symbiotic microbes were also found to influence resource allocation patterns, resulting in plant trait trade-offs [10]. For example, an underground symbiotic bacterial community can favor resource allocation toward growth over pathogen resistance in *Arabidopsis* under suboptimal light [11]. An epiphytic fungus on root surfaces can shift the investment toward growth over herbivore resistance in *Cucumis sativus* [5]. Increasing evidence suggests that growth–defense trade-offs in plants driven by symbiotic microbes mainly involve signaling crosstalk between gibberellin (GA) and other phytohormones, such as salicylic acid (SA) or jasmonic acid (JA) [5,11].

JA-dependent signaling pathways are usually associated with plant herbivore defense, whereas SA-dependent signaling pathways are most often thought to function in plant pathogen defense [12]. In addition, H_2_O_2_ can act as a signaling molecule to participate in the SA-dependent anti-pathogen signaling pathway [13,14]. Previous studies have suggested that JA-dependent herbivore defense signaling is often antagonized by SA-dependent pathogen defense signaling in plants, resulting in a trade-off between pathogen and herbivore defense traits [12]. Interestingly, symbiotic microbes were also found to upregulate SA and JA levels in plants [15,16,17] and drive the antagonism between the two hormonal signaling pathways [18,19]. These results implied that symbiotic microbes might drive a trade-off between SA-dependent pathogen defense and JA-dependent herbivore defense.

Understanding the underlying mechanism involved in a trade-off between plant pathogen and herbivore resistance can help with plant production and management [20]. *Ipomoea cairica* is a destructive invasive liana in South China. This species originates from tropical Africa. It was first documented in South China in 1912 [21]. In the introduced regions, it has evolved multiple excellent traits, including high salt tolerance and overwhelming growth advantage over co-occurring native species [22,23]. Moreover, *I. cairica* has strong resistance to most natural phytopathogens. Only a mild anthracnose symptom on its leaves caused by *Colletotrichum gloeosporioides* was documented in earlier field investigations [24]. Thus, it shows considerably high fitness in the introduced regions and causes serious losses to local agriculture, forestry, and biodiversity [24]. Our previous study already suggested that these excellent traits of *I. cairica* are associated with the symbiosis of two epiphytic fungi, *F. oxysporum* and *F. fujikuroi* [22,23,24]. Several chemical herbicides are effective at controlling *I. cairica*, but their application inevitably causes serious environmental pollution [25]. Thus, biological control may be an attractive approach to control this invasive plant. Intriguingly, our latest investigation in fields has observed that an insect herbivore, *Taiwania circumdata*, often infests many *I. cairica* populations and can cause serious herbivorous damage on its leaves, which means that this plant has low herbivore resistance against the insect herbivore. Therefore, *T. circumdata* may be regarded as a potential biocontrol agent for *I. cairica*. To guide the design of a biocontrol strategy for *I. cairica*, it is essential to understand the underlying mechanisms for its low herbivore resistance, which currently remain unknown.

Based on the ability of symbiotic microbes to drive the antagonism of SA on JA signaling pathways [18], we hypothesized that the mixture of epiphytic *F. oxysporum* and *F. fujikuroi* might induce pathogen resistance of *I. cairica* against *C. gloeosporioides* at the cost of sacrificing its herbivore resistance against *T. circumdata*. Therefore, we aim to answer the following questions: (i) whether the epiphytic fungal mixture can reduce herbivore resistance of *I. cairica* against *T. circumdata*; (ii) if so, whether the reduction in the herbivore defense is coupled to an increase in the pathogen defense in *I. cairica*; and (iii) whether the trade-off between the two defense traits involves antagonism between SA and JA defense signaling in *I. cairica*. Here, superoxide dismutase (SOD), peroxidase (POD), and catalase (CAT) activities were used as indicators to assess generalized defense levels. Disease symptoms, SA level, H_2_O_2_ concentration, and chitinase and β-1,3-glucanase activities were used as indicators to assess pathogen defense levels, while herbivorous damage symptoms, JA level, phenolic compound content, and phenylalanine ammonia-lyase (PAL) activity were used as indicators to assess herbivore defense levels.

## 2. Materials and Methods

### 2.1. Plant Materials

*I. cairica* cuttings were collected from sample sites located in Yakou Village, Zhongshan, China (22°21′ N, 113°30′ E). One hundred *I. cairica* cuttings (18–20 cm length, 2 mm diameter), each with two healthy leaves, were collected from different plant individuals. These cuttings were returned to the laboratory and cultivated individually in 150 mL of sterile Hoagland solution contained in a 350 mL hydroponic tank for two days under conditions of room temperature and natural light.

### 2.2. Insect Materials

*T. circumdata* adults were collected from plant sample sites. Several *I. cairica* populations were infested with *T. circumdata*. A population of *T. circumdata* adults (≈5 mm in length) was captured using a sweeping net. These adults were returned to the laboratory and fed fresh *I. cairica* leaves in clip cages (30 × 30 × 30 cm) for 2 days. Before they were used in experiments, they were starved for 1 day.

### 2.3. Fungal Materials

The isolates of epiphytic fungi *F. oxysporum* and *F. fujikuroi* and the pathogenic fungus *C. gloeosporioides* used here were the collections in our previous study. Original strains of the three fungal species were recovered from leaf surfaces on the same individual *I. cairica* plant grown in the plant sample sites. These original isolates underwent two generations of subculture until axenic cultures were obtained. The axenic cultures of the three fungal species were identified with combined molecular and morphological analysis methods [23]. Before they were used in experiments, these axenic cultures of *F. oxysporum*, *F. fujikuroi,* and *C. gloeosporioides* were selected for subculture on sterile potato dextrose agar (PDA; potato 20%, dextrose 2%, agar 1.5%) in the dark at 26 °C for 14 days.

### 2.4. Preparation of Conidial Suspensions of Epiphytic and Pathogenic Fungi

Following the method of Xu et al. [23], these 14-day-old cultures of *F. oxysporum*, *F. fujikuroi,* and *C. gloeosporioides* were used for the preparation of their respective conidial suspensions. Briefly, fungal mycelia (0.15 g) were scraped from PDA and transferred to 50 mL of sterile potato dextrose broth (PDB; potato 20%, dextrose 2%) contained in a 250 mL sterile flask, which was then sealed with Parafilm^®^ (Shanghai Suolaibao Biotechnology, Shanghai, China) and incubated in darkness at 26 °C for 24 h. The suspension of each fungus was filtrated into a 50 mL sterile centrifuge tube with three layers of sterile gauze to remove mycelia. Each filtrate was centrifuged at 4000 rpm for 10 min to allow conidia to deposit at the bottom of the centrifuge tube, and then the supernatant was discarded. The conidia left in the centrifuge tube were resuspended in 10 mL of sterile PDB. The conidial concentration in the suspension of each fungal species was measured by microscopic observation using a hemocytometer (Shanghai Medical Optical Instrument Plant, Shanghai, China) and then adjusted to 2 × 10^6^ mL^−1^ with sterile PDB.

The epiphytic fungal mixture was prepared by blending conidial suspensions of *F. oxysporum* and *F. fujikuroi* together in equal volumes, yielding a total conidial concentration of 2 × 10^6^ mL^−1^.

### 2.5. Preparation of Sterile Plant Materials

Fifty *I. cairica* cuttings were used for the preparation of sterile cuttings. According to the methods of Xu et al. [23], the surfaces of *I. cairica* cuttings were sterilized by cleaning twice with degreased cotton soaked in 75% (*v*:*v*) ethanol. They were grown in their original Hoagland solution and incubated in sterile phytotrons at 28 °C with a 14 h/10 h light/dark photoperiod (cool-white neon tube, 200 µmol·m^−2^·s^−1^). The relative humidity in the phytotrons was maintained at 85%. Two hours later, these sterile cuttings were used as plant materials for the experiments.

### 2.6. Preparation of Epiphyte-Inoculated and Epiphyte-Free Plant Materials

Twenty sterile cuttings were selected from phytotrons and sprayed with 2.5 mL of the epiphytic fungal mixture, serving as epiphyte-inoculated (E+) cuttings. Another 20 sterile cuttings were selected from phytotrons and sprayed with an equal volume of sterile PDB, serving as epiphyte-free (E−) cuttings. Both epiphytic fungal mixture and sterile PDB were sprayed on leaf surfaces. These E+ and E− cuttings were returned to their original Hoagland solutions to grow. To avoid cross-infection of epiphytic fungi, the E+ and E− groups were incubated separately in their respective sterile phytotrons under the same temperature, light, and humidity conditions described in Section 2.5. Three days later, these E+ and E− cuttings were used as plant materials for experiments.

### 2.7. Assessing Pathogen Resistance of E+ and E− Cuttings After C. gloeosporioides Infection

To investigate the effects of epiphytic fungi on the pathogenic resistance of *I. cairica*, we performed a trial of *C. gloeosporioides* infection of E+ and E− cuttings. Six E+ and six E− cuttings that had been incubated for 3 days as described in Section 2.6 were selected randomly from phytotrons and then sprayed with the conidial suspension of *C. gloeosporioides*. The spray volume was 2 mL per leaf. Then, these E+ and E− cuttings were grown in their original Hoagland solutions and incubated separately in two sterile phytotrons under the same temperature, light, and humidity conditions described in Section 2.5. After 3 days, all cuttings were harvested and their disease symptoms (i.e., rotted leaf and yellowed leaf) recorded. Rotted and yellowed leaf rates per plant were calculated as a percentage of the total leaf number per plant.

### 2.8. Assessing Herbivore Resistance of E+ and E− Cuttings After T. circumdata Introduction

To investigate the effects of epiphytic fungi on herbivore resistance of *I. cairica*, we performed a trial of *T. circumdata* introduction to E+ and E− cuttings. Six E+ and 6 E− cuttings that had been incubated for 3 days as described in Section 2.6 were selected randomly from phytotrons and used in a trial of *T. circumdata* introduction. Here, Petri dishes (diameter 10 cm, height 1 cm) without lids and containing sterile Hoagland solution (30 mL) were sealed by a layer of plastic wrap, serving as modified hydroponic tanks for leaf cultivation. The plastic wrap was punctured by a needle, forming a hole (≈diameter 2 mm) just small enough for the petiole of *I. cairica* to pass through. One leaf with an intact petiole was clipped randomly from each E+ and E− cutting. After weighing, the mass of the leaf was recorded as the original fresh weight. The leaf was then cultivated in a modified hydroponic tank that had been placed in advance in a clip cage (30 × 30 × 30 cm) enclosed with a colorless nylon net (40 mesh). After this, one *T. circumdata* adult was introduced into the clip cage and placed on the leaf. The leaf (E+)–herbivore and leaf (E−)–herbivore pairs contained in clip cages were then incubated separately in two sterile phytotrons under the same temperature, light, and humidity conditions described in Section 2.5. Three days later, all the E+ and E− leaves were harvested. The moisture on their petiole surfaces was blotted up with bibulous paper. This was followed by weighing their residual fresh weights. The degree of herbivorous damage per leaf was assessed using the absolute and relative fresh weight loss. Absolute fresh weight loss per leaf was expressed as the difference between its original and residual fresh weight. Relative fresh weight loss per leaf was expressed as the percentage of the difference to its original fresh weight.

### 2.9. Assessing Defense Responses of E+ and E− Cuttings Before Pathogen Inoculation and Herbivore Introduction

To further investigate the effects of epiphytic fungi on pathogen and herbivore defense responses in *I. cairica*, we performed four trials of epiphytic fungal pre-inoculation of sterile *I. cairica* cuttings without subsequent pathogen inoculation and herbivore introduction. In these trials, plant surface sterilization, epiphytic fungal inoculation, and the preparation of E+ and E− cuttings were carried out using the same methods described in Section 2.5 and Section 2.6. The cuttings in the E+ and E− groups were incubated separately in two sterile phytotrons under the same temperature, light, and humidity conditions described in Section 2.5. Three days later, all of the cuttings were harvested.

In trial 1, E+ and E− cuttings, each with 4 biological replicates, were used for the analysis of pathogen defense enzymes, including chitinase and β-1,3-glucanase activities. In trial 2, E+ and E− cuttings, each with 4 biological replicates, were used for the measurement of herbivore defense enzymes and compounds, including total phenol content and PAL activity. In trial 3, E+ and E− cuttings, each with 5 biological replicates, were used for the analysis of generalized defense enzymes, including SOD, POD, and CAT activities. In trial 4, E+ and E− cuttings, each with 4 biological replicates, were used for the detection of defense signaling molecules, including SA and JA contents and H_2_O_2_ concentration.

### 2.10. Parameter Determination

The extraction of β-1,3-glucanase and chitinase was performed using the method described by Magnin-Robert et al. [26]. Deveined fresh leaves (0.2 g) were homogenized in 5 mL cold sodium acetate buffer (pH 5.0) containing 1 mmol dithiothreitol and 10 mg phenylmethylsulfonyl fluoride. The homogenate was centrifuged at 4000 rpm for 5 min at 4 °C, and the supernatant was used in assays to quantify β-1,3-glucanase and chitinase activities.

β-1,3-glucanase activity was measured using the method developed by De la Cruz et al. [27]. A 200 µL volume of supernatant was mixed with 200 µL of laminarin (1 mg·mL^−1^) and incubated at 37 °C for 30 min, after which 2 mL of dinitrosalicylic acid (DNS) reagent (Sangon Biotech Co., Ltd., Shanghai, China) was added before boiling for 5 min. Enzyme and substrate blanks were run simultaneously. The reaction mixture’s absorbance was recorded at 600 nm. A standard curve was established with 0–80 mg·mL^−1^ glucose. One unit of β-1,3-glucanase activity was defined as the amount of enzyme required to catalyze the release of 1 μmol of glucose equivalent per min.

Chitinase activity was measured using the method outlined by Chen and Lee [28]. A 400 µL volume of supernatant was mixed with 400 µL of colloidal chitin (10 mg·mL^−1^). The mixture was incubated at 37 °C for 1 h, after which 1.5 mL of DNS reagent was added before boiling the mixture for 5 min. Enzyme and substrate blanks were run simultaneously. The reaction mixture’s absorbance was recorded at 530 nm. A standard curve was established with 0–1 mg·mL^−1^ N-acetylglucosamine (NAG). One unit of chitinase activity was defined as the amount of enzyme required to catalyze the release of 0.5 µmol of NAG equivalent per hour.

The extraction of SOD, POD, and CAT was performed according to the method presented by Dhindsa et al. [29]. Deveined fresh leaves (0.5 g) were homogenized in 5 mL of 0.05 mol·L^−1^ phosphate buffer (pH 7.0) in a cold mortar. The homogenate was centrifuged at 5000 rpm for 20 min at 4 °C, after which the supernatant was assayed to determine SOD, POD, and CAT activities.

SOD activity was assessed using a method adapted from Dhindsa et al. [29]. A 3 mL reaction mixture was added to a test tube containing 0.05 mol·L^−1^ phosphate buffer (pH 7.8), 13 mM methionine, 75 μM nitro blue tetrazolium (NBT), 2 μM riboflavin, 0.1 mM EDTA, and 30 μL enzyme extract. An equal volume of heat-killed enzyme extract was mixed with the same substrates in a test tube. A blank was assembled by mixing an equal volume of heat-killed enzyme extract with the same substrates in another test tube sealed with tinfoil. The three test tubes were exposed to light (cool-white neon tube, 170 µmol·m^−2^·s^−1^) for 20 min, after which the absorbance of the reaction mixtures was recorded at 560 nm. One unit of SOD activity was defined as the inhibition of 50% of the initial photochemical reduction of NBT under illumination.

POD activity was assessed using a method adapted from Ge et al. [30]. A 100 µL volume of heat-killed enzyme extract was mixed with 1 mL of 0.3% (m:m) H_2_O_2_ aqueous solution, 1 mL of 0.2% (v:v) guaiacol in 0.05 mol·L^−1^ phosphate buffer (pH 7.0), and 1 mL of 0.05 mol·L^−1^ phosphate buffer (pH 7.0) in a cuvette as a blank. An equal volume of enzyme extract was mixed rapidly with the same substrates in another cuvette, and then the change in absorbance of the reaction mixture was recorded at 470 nm over 1 min. One unit of POD activity was defined as an increase in absorbance of 0.01 per minute.

CAT activity was measured by adapting the method described by Dhindsa et al. [29]. A 100 µL volume of heat-killed enzyme extract was mixed with 2 mL of 0.05 mol·L^−1^ phosphate buffer (pH 7.0) and 1 mL of 0.3% (m:m) H_2_O_2_ in a cuvette as a blank. An equal volume of enzyme extract was mixed rapidly with the same substrates in another cuvette, after which the change in absorbance of the reaction mixture was recorded at 240 nm over 1 min. One unit of CAT activity was defined as a reduction in absorbance by 0.01 per min.

MDA content was measured by adapting the method described by Zhang et al. [31]. Deveined fresh leaves (0.2 g) were homogenized in 10 mL of 10% trichloroacetic acid and centrifuged at 4000 rpm. Then, 2 mL of the supernatant was transferred to a test tube using a pipette, where it was mixed with 2 mL of 0.6% thiobarbituric acid. The mixture was heated at 100 °C for 15 min, cooled quickly, then centrifuged at 4000 rpm for 10 min. The absorbance of the mixture was recorded at 600, 532, and 450 nm with an ultraviolet spectrophotometer. MDA content (µmol·g^−1^ FW) was calculated using the method outlined by Zhang et al. [31].

The extraction of PAL and its activity assay were performed following a method adapted from Sallam et al. [32]. Deveined fresh leaves (0.1 g) were homogenized in 5 mL of 0.1 mol·L^−1^ sodium borate buffer (pH 8.8) containing 2 mmol·L^−1^ mercaptoethanol. The homogenate was centrifuged at 12,000 rpm for 20 min at 4 °C. Then, the supernatant was assayed to quantify PAL activity. The reaction mixture contained 0.5 mL of the supernatant, 3.5 mL of 0.1 mol·L^−1^ sodium borate buffer (pH 8.8), and 1 mL of 0.02 mol·L^−1^ L-phenylalanine. The mixture was incubated at 40 °C for 30 min. The enzymatic reaction was halted by adding 0.2 mL of 6 mol·L^−1^ hydrochloric acid. A blank was constructed by replacing the supernatant in the reaction mixture with 0.5 mL of 0.1 mol·L^−1^ borate buffer (pH 8.8). The reaction mixture’s absorbance was recorded at 290 nm. One unit of PAL activity was defined as an increase in absorbance of 0.01 per min.

The extraction and measurement of total phenol was performed using a method adapted from Marzorati et al. [33]. Deveined fresh leaves (0.1 g) were homogenized in 4 mL ethanol and 0.4 mL 10% (*v*:*v*) trichloroacetic acid. The homogenate was transferred into a 10 mL centrifuge tube and diluted to a final volume of 10 mL by adding 10% trichloroacetic acid, after which the mixture was centrifuged at 4000 rpm for 5 min. The supernatant was then assayed to measure the total phenol content. The reaction mixture contained 0.5 mL of supernatant, 2 mL of 15% (*m*:*v*) Na_2_CO_3_ aqueous solution, 2 mL of 0.2 mol·L^−1^ Folin–Ciocalteu reagent (Sangon Biotech Co., Ltd., Shanghai, China), and 5.5 mL of distilled water. The mixture was incubated at 25 °C for 1 h. To assemble a blank, the supernatant in the reaction mixture was replaced by 0.5 mL of distilled water. The absorbance of the reaction mixture was recorded at 765 nm. A standard curve was established with 0–70 μg·mL^−1^ gallic acid. Total phenol content was expressed in terms of gallic acid equivalents (μg·g^−1^ FW).

SA and JA were extracted and qualified according to the method of Xu et al. [24]. Frozen leaves (1.0 g) were ground in liquid nitrogen into a fine powder. Extraction was performed by adding 10 mL of cold methanol. The homogenate was transferred to a 50 mL centrifuge and then set on a shaker at 300 rpm for 2 h. After centrifugation at 4000 rpm for 5 min, the supernatant was transferred to another centrifuge tube and concentrated under a flow of nitrogen gas. The residue was reconstituted with 1 mL of cold methanol and then filtered through a 0.2 μm Teflon membrane filter into an autosampler vial. An AB Sciex Qtrap^®^ 5500 LC/MS/MS system (AB Sciex, Foster City, CA, USA) with multiple reaction monitoring modes was used for the quantification of SA and JA. The sample was injected into a reverse-phase column PAKC18-ARC (150 × 20 mm, 3 μm, Shiseido, Tokyo, Japan) kept at 25 °C and eluted isocratically with the mobile phase consisting of 5 mM ammonium acetate (mobile phase A) and acetonitrile (mobile phase B) at a flow rate of 0.3 mL min^−1^. The injection volume was 0.2 μL. The elute was subjected to positive electrospray ionization and the ions were detected using the following mass transitions: SA *m*/*z* 137.0 → *m*/*z* 93; JA *m*/*z* 209.0 → *m*/*z* 59.0. The external standard working fluids for calibration curves were established with 2–100 ng mL^−1^ of SA and JA in methanol. The standards for SA and JA were purchased from ZZBIO Co., Ltd. (Shanghai, China).

H_2_O_2_ extraction from leaves was based on the method outlined by Ferguson et al. [34]. Deveined fresh leaves (0.2 g) were homogenized in 5 mL of cold acetone. The homogenate was then centrifuged at 4000 rpm for 10 min at 4 °C. H_2_O_2_ was measured in the supernatant using a method adapted from Brennan and Frenkel [35]. A 1 mL volume of supernatant was thoroughly mixed with 250 µL of 50 mg·L^−1^ Ti(SO_4_)_2_ in concentrated H_2_SO_4_, after which 2 mL of concentrated NH_4_OH was added. After centrifugation at 4000 rpm for 20 min, the supernatant was discarded. The precipitate was washed repeatedly with 4 mL acetone until the supernatant was colorless, after which it was dissolved in 4 mL of 1 mol·L^−1^ H_2_SO_4_. The solution’s absorbance was recorded at 415 nm against a water blank. The concentration of H_2_O_2_ in the extract was determined by comparing the absorbance against a standard curve representing 0–80 µmol·L^−1^ H_2_O_2_.

### 2.11. Statistical Analysis

Statistical analyses were performed by SPSS 16.0 software (IBM, Chicago, IL, USA) using one-way analysis of variance (ANOVA) followed by independent sample *t*-test and LSD test at 5% and 1% significance levels. Values were expressed as means ± standard errors.

## 3. Results

### 3.1. Disease Symptoms of E+ and E− I. cairica After C. gloeosporioides Infection

After the infection by *C. gloeosporioides*, there was a significant difference in the rotted (*p* = 0.038, Figure 1a) and yellowed leaf (*p* = 0.047, Figure 1b) rates between E+ and E− cuttings (Figure 1). Both the rotted and yellowed leaf rates in E− cuttings were 66.7 ± 25.8% and 66.7 ± 10.5%. However, the two rates in E+ cuttings were 0.0 ± 0.0% and 8.3 ± 8.3%, respectively. These results suggest that E+ cuttings had higher pathogen resistance than E− cuttings.

### 3.2. Degrees of Herbivorous Damage of E+ and E− I. cairica After T. circumdata Introduction

After the introduction of *T. circumdata*, both relative (*p* = 0.014, Figure 2a) and absolute (*p* = 0.009, Figure 2b) fresh weight losses per leaf of E+ cuttings were significantly lower than those of E− cuttings. These results suggest that E+ cuttings had lower herbivore resistance than E− cuttings.

### 3.3. Enzyme Activities Associated with Pathogen Defense in E+ and E− Cuttings Before Natural Enemy Introduction

Before *C. gloeosporioides* inoculation and *T. circumdata* introduction, E+ cuttings had higher chitinase (*p* = 0.002, Figure 3a) and β-1,3-glucanase activities (*p* = 0.001, Figure 3b) than E− cuttings. This result suggests that the mixture of *F. fujikuroi* and *F. oxysporum* had already induced pathogen defense in *I. cairica* cuttings before the introduction of the two natural enemies.

### 3.4. Enzyme Activities and Compound Content Associated with Herbivore Defense in E+ and E− Cuttings Before Natural Enemy Introduction

Before *C. gloeosporioides* inoculation and *T. circumdata* introduction, PAL activity (*p* = 0.010, Figure 4a) and total phenol content (*p* = 0.011, Figure 4b) in E+ cuttings were significantly lower than those in E− cuttings. This result suggests that the mixture of *F. fujikuroi* and *F. oxysporum* had already inhibited herbivore defense in *I. cairica* cuttings before the introduction of the two natural enemies.

### 3.5. Enzyme Activities Associated with Generalized Defense in E+ and E− Cuttings Before Natural Enemy Introduction

Before *C. gloeosporioides* inoculation and *T. circumdata* introduction, there were no significant differences in SOD (*p* = 0.574, Figure 5a), POD (*p* = 0.621, Figure 5b), and CAT (*p* = 0.754, Figure 5c) activities between E+ and E− cuttings. This result suggests that the mixture of *F. fujikuroi* and *F. oxysporum* could not trigger generalized defense in *I. cairica* cuttings before the introduction of the two natural enemies.

### 3.6. Signaling Molecule Levels Associated with Pathogen and Herbivore Defense in I. cairica Cuttings Before Natural Enemy Introduction

Before *C. gloeosporioides* inoculation and *T. circumdata* introduction, SA content (*p* = 0.016, Figure 6a), JA content (*p* = 0.048, Figure 6b), and H_2_O_2_ concentration (*p* = 0.049, Figure 6c) in E+ cuttings were significantly higher than those in E− cuttings. This result suggests that the mixture of *F. fujikuroi* and *F. oxysporum* had already elicited the accumulation of these defense signaling molecules in *I. cairica* cuttings before the introduction of the two natural enemies.

## 4. Discussion

### 4.1. Roles of Epiphytes in Driving Trade-Offs Between Defense Traits in I. cairica

Trade-offs between different plant traits exist commonly in plants and are regarded to be a result of species evolution [2]. For example, *Ambrosia artemisiifolia* evolves drought tolerance at the cost of herbivore tolerance in its introduced regions [36]. *Arabidopsis* evolves defense responses to a bacterial pathogen at the cost of growth [37]. A rice mutant evolves pathogen resistance, resulting in reduced thermo-tolerance [8]. Additionally, a trade-off between pathogen and herbivore resistance was reported in many plant species, including *Zea mays* [20], *Phaseolus lunatus* [38,39], and *Solanum lycopersicum* [40]. *I. cairica* has experienced a century of invasion history and evolved salt tolerance, pathogen resistance, and rapid clonal reproduction in South China [22,23,24]. A field survey conducted earlier in Fujian Province, Southeast China, documented a severe herbivorous damage symptom caused by *T. circumdata* in many *I. cairica* populations [41]. A similar herbivorous damage symptom was also observed in the present study (Figure 2a,b). These results suggest that *I. cairica* has a low herbivore resistance against *T. circumdata*. A previous study showed that invasive *A. artemisiifolia* evolved drought tolerance at the cost of sacrificing herbivore resistance [36]. Therefore, we inferred that the low herbivore resistance of *I. cairica* might be the result of trade-offs with its several excellent traits, such as rapid vegetative growth, strong pathogen resistance, and high salt tolerance.

In recent years, researchers have begun to pay attention to the potential role of beneficial symbiotic microbes in driving plant trait trade-offs [5,10,11]. These plant-associated microbes survive by consuming carbohydrates and amino acids provided by host plants [42,43]. In return, they can confer multiple benefits to host plants [44,45]. These benefits include promoting plant growth and inducing plant tolerance to biotic and abiotic stress [46,47,48,49,50,51,52]. Still, no direct evidence suggests that symbiotic microbes are able to eliminate or alleviate trade-offs between multiple plant traits [10]. On the contrary, symbiotic microbial species and communities were found to drive growth–defense trade-offs in plants [5,11], which provided a novel clue for screening potential biotic factors associated with the trade-off between pathogen and herbivore resistance in plants.

Our earlier studies have shown that epiphytic *F. oxysporum* and *F. fujikuroi* have established a symbiotic relationship with many *I. cairica* populations in fields and conferred multiple benefits to them, including pathogen resistance [22,23,24]. In the present study, high pathogen resistance was accompanied by low herbivore resistance in E+ cuttings (Figure 1a,b and Figure 2a,b). This result suggests that the epiphytic fungal mixture could serve as a crucial biotic factor driving a trade-off between pathogen and herbivore defenses in *I. cairica*.

### 4.2. Mechanisms by Which Epiphytes Drive Trade-Offs Between Defense Traits in I. cairica

Signaling crosstalk among various hormonal molecules regulates the final performance of plants, including growth, survival, and defense [53]. Due to the ability to manipulate hormonal signaling crosstalk, symbiotic microbes are believed to affect plant performance [54]. Previous studies have suggested that symbiotic microbes can drive growth–defense trade-offs in plants, which mainly involves signaling crosstalk between GA and other phytohormones such as SA or JA [5,11]. Interestingly, many studies have shown that symbiotic microbes can elicit the antagonism between SA and JA signaling in plants [18,19], resulting in the interruption of downstream signaling transmission of SA or/and JA [24].

SA primarily activates plant pathogen defenses through signaling transduction [11], which typically involves the activation of chitinase and β-1,3-glucanase [55]. These two enzymes can degrade chitin and glucan, respectively, in the cell walls of pathogenic fungi [56,57]. Thus, they are regarded as important enzymes associated with plant pathogen defense. In addition, H_2_O_2_ usually acts as a signal molecule in SA-dependent anti-pathogen signaling pathways [13,14]. Many studies have suggested that symbiotic microbes possess the ability to induce the accumulation of SA and H_2_O_2_ in plants [13]. A similar result was observed in the present study, as evidenced by higher SA and H_2_O_2_ levels in E+ cuttings than in E− cuttings (Figure 6a,c), which suggests that the epiphytic fungal mixture can induce synthesis of the two signaling molecules in *I. cairica* before *C. gloeosporioides* inoculation. Consistently, higher chitinase and β-1,3-glucanase activities were also observed in E+ cuttings (Figure 3a,b), which suggested that SA-dependent pathogen defense signaling in E+ cuttings had been successfully transmitted to downstream pathways before *C. gloeosporioides* inoculation, coinciding with its milder disease symptoms after *C. gloeosporioides* infection (Figure 1a,b).

JA primarily elicits plant herbivore defense pathways [5,11], which typically involve the activation of PAL and the accumulation of phenolic compounds [55,58,59]. Phenolic compounds and PAL are involved in phytoalexin synthesis in plants [14]. Thus, total phenol content and PAL activity are regarded as indicators for assessing plant herbivore resistance. Earlier studies suggested that endophytic and mycorrhizal fungi could elicit JA accumulation and activate JA-dependent herbivore defense responses in plants [19,59]. Interestingly, JA-dependent signaling pathways are often antagonized by SA signaling [60]. In this case, the JA-dependent herbivore defense may be inhibited. In the present study, higher JA levels occurred in E+ cuttings (Figure 6b), which confirmed that the epiphytic fungal mixture could also induce the synthesis of JA in *I. cairica* before *T. circumdata* introduction. However, the higher JA level in E+ cuttings was accompanied by lower PAL activity and total phenolic content (Figure 4a,b). This result suggests that JA-dependent herbivore defense signaling in E+ cuttings failed to be successfully transmitted to downstream pathways before *T. circumdata* introduction, which coincided with its severe herbivorous damage symptoms after *T. circumdata* infestation (Figure 2a,b). Thus, we inferred that JA-dependent herbivore defense responses in *I. cairica* might be antagonized by SA signaling due to the symbiosis of the epiphytic fungal mixture.

Earlier studies suggested that SA and JA can also activate several generalized defense enzymes associated with anti-pathogens and anti-herbivores in plants, such as SOD, POD, and CAT [58,61]. However, a similar result was not observed in the present study (Figure 5a–c), which suggests that generalized defense signaling pathways regulated by the two hormonal molecules failed to be activated in *I. cairica* by the epiphytic fungal mixture. We infer that this might involve more complex signaling crosstalk among various hormones.

### 4.3. Application of the Theory of Plant Trait Trade-offs in Biological Control of I. cairica

Artificially driving or alleviating plant trait trade-offs is of great application value in plant production [62,63,64]. Studies have suggested that if crops are grown for yield purposes, cultivation regimes adopted should fine-tune the allocation of more photosynthates toward seed rather than vegetative parts [63]. For example, exogenous application of plant growth regulators, such as ethephon and cycocel, was shown to reduce internode length and facilitate assimilate partitioning to the ear in maize, which in turn increases the final yield [65]. Likewise, the theory of plant trait trade-offs can be applied to control invasive plants. For example, invasive *A. artemisiifolia* evolved drought tolerance at the cost of sacrificing herbivore resistance in introduced regions. In this case, natural enemy introduction is considered an effective approach for controlling this plant [36].

Our previous study showed that the strong pathogen resistance of *I. cairica* against *C. gloeosporioides* is associated with the symbiosis of epiphytic *F. oxysporum* and *F. fujikuroi* [24]. Thus, we originally believed that *C. gloeosporioides* could be used as a biocontrol agent for *I. cairica* as long as these two epiphytic fungi were removed from the plant in advance [24]. However, the removal of these epiphytic fungi requires the use of chemical fungicides, which will inevitably cause serious environmental pollution issues. Clearly, this is not an environmentally friendly control strategy. In the present study, the pre-inoculation of an epiphytic fungal mixture enhanced the pathogen resistance of *I. cairica* against *C. gloeosporioides* at the cost of sacrificing its herbivore resistance against *T. circumdata* (Figure 1a,b and Figure 2a,b). This means that *T. circumdata* is a potential biocontrol agent for *I. cairica*. In this case, biocontrol strategies for *I. cairica* should be based on maintaining symbiotic relationships with the two epiphytic fungi rather than destroying them. Thus, we believed that maintaining or artificially establishing symbiosis between *I. cairica* and the two epiphytic fungi can reduce the herbivore defense of this plant, thus facilitating the control of this plant by *T. circumdata*. Notably, the biosafety of *T. circumdata* to co-occurring native plant species of *I. cairica* should be evaluated comprehensively before its introduction.

## 5. Conclusions

The mixture of epiphytic *F. oxysporum* and *F. fujikuroi* can drive a trade-off between herbivore and pathogen defenses in *I. cairica*, resulting in reduced resistance against *T. circumdata* but enhanced resistance against *C. gloeosporioides* in this plant. The trade-off between the two defense traits in *I. cairica* driven by the epiphytic fungal mixture involves the antagonism of SA on JA signaling. Therefore, maintaining or establishing a symbiotic relationship between *I. cairica* and the two fungi in fields may facilitate the control of this plant by *T. circumdata*. This study supplements evidence that epiphytic fungi can drive a trade-off between plant pathogen and herbivore defenses and provides a novel insight into biocontrol strategies for invasive *I. cairica.*

## Figures and Tables

**Figure 1 microorganisms-12-02130-f001:**
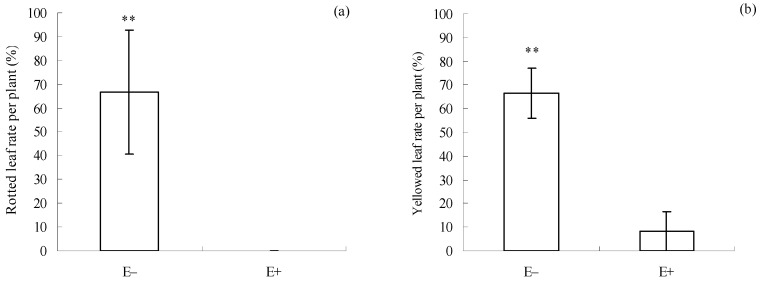
Disease symptoms of E+ and E− *I. cairica* after *C. gloeosporioides* infection. (**a**) Rotted leaf rate per plant. (**b**) Yellowed leaf rate per plant. E− and E+ indicate *I. cairica* cuttings inoculated with sterile PDB, and the conidial mixture of *F. fujikuroi* and *F. oxysporum*, respectively. Each value is presented as the mean ± standard error of six biological replicates per treatment. ** indicates a significant difference as determined by independent sample *t*-test at a 1% significance level.

**Figure 2 microorganisms-12-02130-f002:**
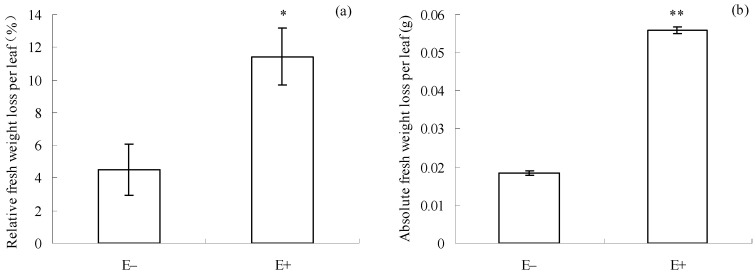
Degree of herbivorous damage of E+ and E− *I. cairica* after *T. circumdata* introduction. (**a**) Relative fresh weight loss per leaf. (**b**) Absolute fresh weight losses per leaf. E− and E+ indicate *I. cairica* cuttings inoculated with sterile PDB, and the conidial mixture of *F. fujikuroi* and *F. oxysporum*, respectively. Each value is presented as the mean ± standard error of six biological replicates per treatment. Error bars indicate standard errors. * and ** indicate significant differences as determined by independent sample *t*-tests at 5% and 1% significance levels, respectively. FW: fresh weight.

**Figure 3 microorganisms-12-02130-f003:**
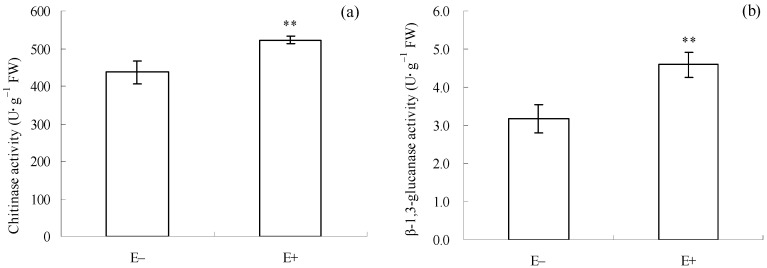
Enzyme activities associated with pathogen defense in *I. cairica* before *C. gloeosporioides* inoculation and *T. circumdata* introduction. (**a**) Chitinase activity. (**b**) β-1,3-glucanase activity. E− and E+ indicate *I. cairica* cuttings inoculated with sterile PDB, and the conidial mixture of *F. fujikuroi* and *F. oxysporum*, respectively. Each value is presented as the mean ± standard error of four biological replicates per treatment. Error bars indicate standard errors. ** indicates a significant difference as determined by independent sample *t*-test at a 1% significance level. FW: fresh weight.

**Figure 4 microorganisms-12-02130-f004:**
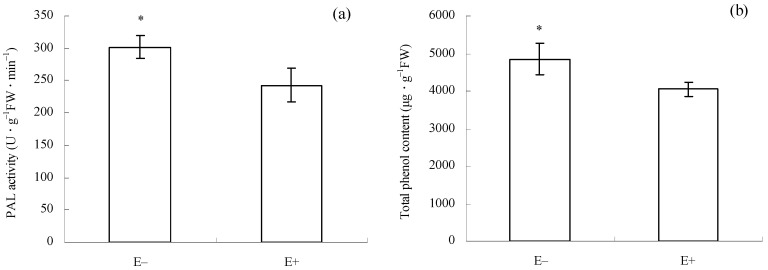
Enzyme activities and compound content associated with herbivore defense in E+ and E− cuttings before *C. gloeosporioides* inoculation and *T. circumdata* introduction. (**a**) Phenylalanine ammonia-lyase (PAL) activity; (**b**) total phenol content. E− and E+ indicate *I. cairica* cuttings inoculated with sterile PDB, and the conidial mixture of *F. fujikuroi* and *F. oxysporum*, respectively. Each value is presented as the mean ± standard error of four biological replicates per treatment. Error bars indicate standard errors. * indicates a significant difference as determined by independent sample *t*-test at a 5% significance level. FW: fresh weight.

**Figure 5 microorganisms-12-02130-f005:**
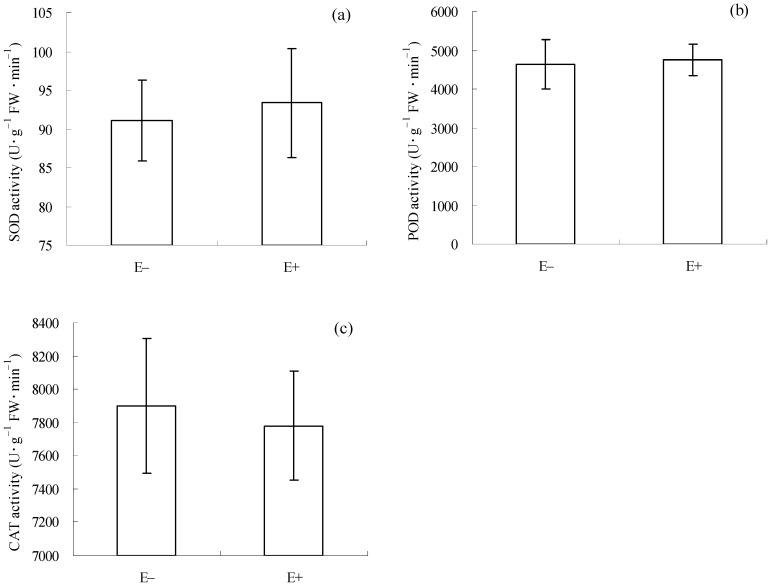
Enzyme activities associated with generalized defense in *I. cairica* before *C. gloeosporioides* inoculation and *T. circumdata* introduction. (**a**) SOD activity; (**b**) POD activity; (**c**) CAT activity. E− and E+ indicate *I. cairica* cuttings inoculated with sterile PDB, and the conidial mixture of *F. fujikuroi* and *F. oxysporum*, respectively. Each value is presented as mean ± standard error of five biological replicates per treatment. Error bars indicate standard errors. FW: fresh weight.

**Figure 6 microorganisms-12-02130-f006:**
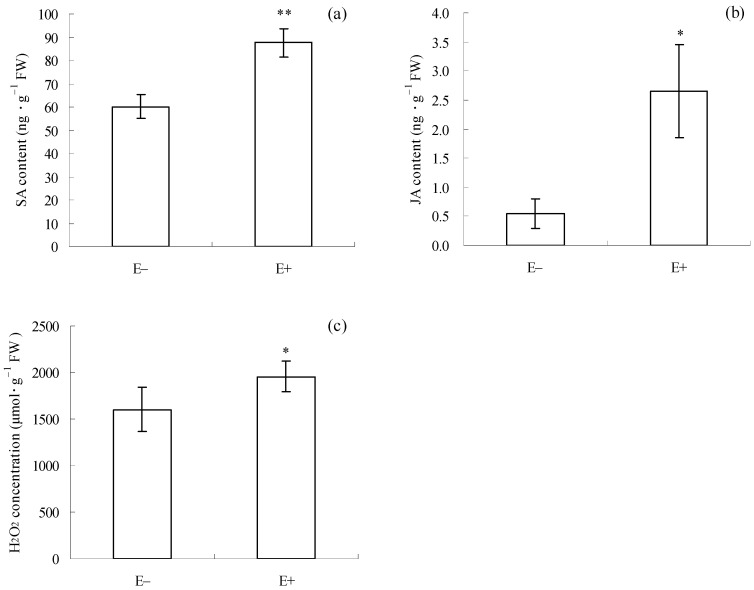
Signaling molecule levels associated with pathogen and herbivore defense in *I. cairica* cuttings before *C. gloeosporioides* inoculation and *T. circumdata* introduction. (**a**) SA content; (**b**) JA content; (**c**) H_2_O_2_ concentration. E− and E+ indicate *I. cairica* cuttings inoculated with sterile PDB, and the conidial mixture of *F. fujikuroi* and *F. oxysporum*, respectively. Each value is presented as the mean ± standard error of four biological replicates per treatment. Error bars indicate standard errors. * and ** indicate significant differences as determined by independent sample *t*-tests at 5% and 1% significance levels, respectively. FW: fresh weight.

## Data Availability

Data are contained within the article.

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
