# Peer review of "Epiphytic Fungi Can Drive a Trade-Off Between Pathogen and Herbivore Resistance in Invasive Ipomoea cairica"

_microorganisms, 2024, doi:10.3390/microorganisms12112130_

Round 1

Reviewer 1 Report

Comments and Suggestions for Authors

In this paper,  authors  investigated whether the mixture of epiphytic fungi Fusarium oxysporum and Fusarium fujikuroi can drive the trade-off between the two defense traits in invasive Ipomoea cairica. As authors stated, no evidence that symbiotic microbes can drive a trade-off between plant pathogen and herbivore defense was presented in the literature, which his the novelty of the paper.

The abstract is too long; please shorten it. The introduction is well documented and makes it easy to understand the state of the art in this area. I have some minor comments.

 It is correct? :to biotic and biotic stress. Pg 2, line 50

Wrong abbreviations: jasmonic acid (SA) or salicylic acid (JA); Pg 2, line 60.

Also, check if the text is in concordance with the chemical names or with abbreviations.

The experimental part is very detailed and easy to follow step by step what the authors done.

Results and discussions are clear and in concordance with the scope of this paper.

Conclusions are not emphasized.

The paper  can be accepted for publication after corrections.

Author Response

Comments 1: [In this paper, authors investigated whether the mixture of epiphytic fungi Fusarium oxysporum and Fusarium fujikuroi can drive the trade-off between the two defense traits in invasive Ipomoea cairica. As authors stated, no evidence that symbiotic microbes can drive a trade-off between plant pathogen and herbivore defense was presented in the literature, which his the novelty of the paper.

The abstract is too long; please shorten it. The introduction is well documented and makes it easy to understand the state of the art in this area. I have some minor comments.

It is correct? :to biotic and biotic stress. Pg 2, line 50

Wrong abbreviations: jasmonic acid (SA) or salicylic acid (JA); Pg 2, line 60.

Also, check if the text is in concordance with the chemical names or with abbreviations.

The experimental part is very detailed and easy to follow step by step what the authors done.

Results and discussions are clear and in concordance with the scope of this paper.

Conclusions are not emphasized.

The paper can be accepted for publication after corrections.]

Response 1:

  1. The abstract is too long; please shorten it.

We agree with this comment. Therefore, we have tried our best to shorten the abstract from 23 lines to 20 lines (Page 1, line 11-30). The main changes are highlighted in red (Page 1, line 17-20).

  1. It is correct? :to abiotic and biotic stress. Pg 2, line 50.

Thank you for pointing this out. We have corrected “responses to abiotic and biotic stress” to “tolerance to biotic and abiotic stress” (Page 1, line 43).

  1. Wrong abbreviations: jasmonic acid (SA) or salicylic acid (JA); Pg 2, line 60. Also, check if the text is in concordance with the chemical names or with abbreviations.

Thank you for pointing this out. We have corrected these mistakes and checked the whole MS for such mistakes (Page 2, line 52).

  1. Conclusions are not emphasized.

We agree with this comment. Thus, we have made the last paragraph of this manuscript a separate conclusion section (Page 12, line 510). 

Reviewer 2 Report

Comments and Suggestions for Authors

This manuscript brings very interesting and valuable information because of its contribution of understanding the mechanisms involved in a trade-off between plant pathogen and herbivore resistance, which can help in agriculture and forestry.

The manuscript is well written (easy to read, excellent English). It is well structured; numerous subchapters provide traceability and understanding of the performance and discovery of this complex research.

I suggest only minor corrections:

Line 60 – Please correct this mistake: instead of “jasmonic acid (SA) or salicylic acid (JA)” you should write “salicylic acid (SA) or jasmonic acid (JA)“.

Line 84-85: Please insert the reference after “our latest investigation in fields …”

Lines 71-90: This whole part (since it is crucial for this study) should be moved to the beginning of the introduction, e.g. after the first three sentences (line 42). The part related to Ambrosia artemisiifolia, Arabidopsis, Zea mays, Phaseolus lunatus and Solanum lycopersicum (lines 42-48) might be omitted here and used in the Discussion.

I noticed „T. Circumdata“ (instead of T. circumdata) in eleven places in the Discussion. Please correct that and check the whole MS for such mistakes.  

Author Response

Comments 2: [This manuscript brings very interesting and valuable information because of its contribution of understanding the mechanisms involved in a trade-off between plant pathogen and herbivore resistance, which can help in agriculture and forestry. The manuscript is well written (easy to read, excellent English). It is well structured; numerous subchapters provide traceability and understanding of the performance and discovery of this complex research.

I suggest only minor corrections:

Line 60 – Please correct this mistake: instead of “jasmonic acid (SA) or salicylic acid (JA)” you should write “salicylic acid (SA) or jasmonic acid (JA)”.

Line 84-85: Please insert the reference after “our latest investigation in fields …”

Lines 71-90: This whole part (since it is crucial for this study) should be moved to the beginning of the introduction, e.g. after the first three sentences (line 42). The part related to Ambrosia artemisiifolia, Arabidopsis, Zea mays, Phaseolus lunatus and Solanum lycopersicum (lines 42-48) might be omitted here and used in the Discussion.

I noticed, T. Circumdata “(instead of T. circumdata)” in eleven places in the Discussion. Please correct that and check the whole MS for such mistakes.]

Response 2:

  1. Line 60: Please correct this mistake: instead of “jasmonic acid (SA) or salicylic acid (JA)” you should write “salicylic acid (SA) or jasmonic acid (JA)”.

Thank you for pointing this out. We have corrected these mistakes (Page 2, line 52).

  1. Line 84-85: Please insert the reference after “our latest investigation in fields …”

Here, we are only describing a phenomenon observed during our recent field investigation, which is not our previously published paper. Thus, there are no references available for insertion here (Page, line 76-78).

  1. Lines 71-90: This whole part (since it is crucial for this study) should be moved to the beginning of the introduction, e.g. after the first three sentences (line 42).

In the introduction section of our manuscript, we first introduced the universality of trait trade-offs in plants, and then discussed the role of symbiotic fungi in driving plant trait trade-offs and the hormonal signaling cross-talk mechanisms involved. Afterwards, we introduced the role of two epiphytic fungi (F. oxysporum and F. fujikuroi) in conferring multiple benefits to I. cairica. These benefits include promoting its growth, and inducing its salt tolerance and pathogen resistance. Based on this logic, we have proposed a hypothesis that the mixture of epiphytic F. oxysporum and F. fujikuroi might induced pathogen resistance of I. cairica against C. gloeosporioides at the cost of sacrificing its herbivore resistance against T. circumdata. If the whole part (Page 2, lines 63-82) is moved to the beginning of the introduce, e.g. after the first three sentences, the introduction section will become illogical. Therefore, we believe that the whole part (Page 2, lines 63-82) should not be moved to other places.

  1. The part related to Ambrosia artemisiifolia, Arabidopsis, Zea mays, Phaseolus lunatus and Solanum lycopersicum (lines 42-48) might be omitted here and used in the Discussion.

We agree with this comment. Thus, we have moved these sentences to the discussion section (Page 10-11, line 407-412).

  1. I noticed, T. Circumdata “(instead of T. circumdata)” in eleven places in the Discussion. Please correct that and check the whole MS for such mistakes.

Thank you for pointing this out. We have corrected these mistakes and checked the whole MS for such mistakes.